# Transformational Leadership and Employees’ Psychological Wellbeing: A Longitudinal Study

**DOI:** 10.3390/ijerph20010676

**Published:** 2022-12-30

**Authors:** Lara Lindert, Sabrina Zeike, Kyung-Eun (Anna) Choi, Holger Pfaff

**Affiliations:** 1Institute of Medical Sociology, Health Services Research and Rehabilitation Science, Faculty of Medicine, University Hospital Cologne, Faculty of Human Sciences, University of Cologne, Eupener Str. 129, 50933 Cologne, Germany; 2Center for Health Services Research, Brandenburg Medical School Theodor Fontane, Fehrbelliner Str. 38, 16816 Neuruppin, Germany; 3vivalue Health Consulting GmbH, Friesenplatz 4, 50672 Cologne, Germany; 4Health Services Research, MIAAI, Danube Private University (DPU) GmbH, Steiner Landstraße 124, 3500 Krems an der Donau, Austria

**Keywords:** social capital, leadership research, mental health, mental risk assessment

## Abstract

Managers play a key role in realizing a humane organization of work. Transformational leadership aims to identify and examine leadership behaviors that strengthen employees’ awareness of the importance and values of task outcomes by articulating a vision for the future, providing a realistic action plan, and giving individualized support. Previous studies have revealed associations between transformational leadership and the psychological wellbeing of employees in different settings, while others did not find such associations. As research based on longitudinal data remains rare, this study builds on longitudinal data from two employee surveys conducted in 2015 and 2018 in a medium-sized German company. In this study, transformational leadership_t0_ and gender had a significant impact on transformational leadership_t1_, while psychological wellbeing_∆_, social capital_∆_, and age did not. Psychological wellbeing_t0_ and social capital_∆_ had a significant impact on psychological wellbeing_t1_, but transformational leadership_∆_, age, and gender did not. Therefore, it is worthwhile for companies to invest in social capital and focus on gender aspects at work. As underlying mechanisms regarding employees’ psychological wellbeing may differ between companies, it is worthwhile for each organization to conduct mental risk assessments to identify “red flags” and implement suitable measures.

## 1. Introduction

Work has many beneficial aspects, including regular income, social participation, and experiencing a sense of purpose. However, working life can also provoke anxiety and perceived threats such as bullying, pack behavior and rank fights, unpleasant external control (e.g., by computer monitoring and managers), high performance requirements and evaluations, uncontrollable changes, and job insecurity, as well as accident risks and health hazards [1]. It is already well examined that work factors have an impact on the psychological wellbeing of employees [2,3,4]. Nonetheless, mental illnesses are still the second commonest reason for absence of work days [5]. Particularly against the backdrop of demographic change and the associated shortage of skilled workers, the mental health of employees is crucial from an economic point of view: it is becoming increasingly important for companies to maintain the health of employees and employ workers for as long as possible [6]. Although this need is acknowledged, many companies—notably small and medium-sized enterprises (SME)—still struggle with investing in employees’ wellbeing (SME) [7]. This is despite the fact that SMEs are particularly affected by demographic change, due to issues such as missing resources (e.g., for demographic management strategies) [8]. Regarding limited financial possibilities, it is vital for SMEs to use the available money in the right place. The job demands-resources model (JD-R model) [9] provides a conceptual framework for positively or negatively associating work aspects with employees’ psychological wellbeing.

### 1.1. Psychological Wellbeing

*“Mental health is a state of mental well-being that enables people to cope with the stresses of life, to realize their abilities, to learn well and work well, and to contribute to their communities”* [10]. The JD-R model [9] is widely used by researchers (e.g., [11,12,13,14,15,16,17,18,19]) and has helped eliminate some of the limitations of previous theoretical models (e.g., Karasek’s job-demand-control model [20]) [21]. The authors of a meta-analytic review based on longitudinal studies concluded that the JD-R model provides a valuable theoretical framework regarding employees’ wellbeing [22].

According to the model, there are several resources and demands that can influence employees’ psychological wellbeing [9]. One of them is the social environment, including social relations at work and leadership, such as social capital or social support from colleagues and supervisors (e.g., information provided by supervisor, high quality relationship with supervisor, constructive feedback) [2,9]. The role of managers in shaping cooperation with employees, providing feedback on work and performance, transparently communicating organizational decisions, and creating opportunities for participation are among the crucial elements in a humane organization of work. Leadership style may function either as a resource or a stressor [2].

### 1.2. Transformational Leadership

Northouse [23] defines leadership as *“a process whereby an individual influences a group of individuals to achieve a common goal.”* Traditional views on leadership effectiveness in the 70s and 80s mainly focused on transactional leadership behavior [24]. Leaders who follow a transactional leadership style focus on employees’ performance (task completion, compliance) and reward or discipline them in this regard [25]. In the late 80s, transformational leadership has moved in the focus of science [24]. Transformational leadership identifies and examines leadership behaviors that strengthen employees’ awareness of the importance and values of task outcomes by articulating a vision for the future, providing a realistic action plan to reach necessary goals, and giving individualized support to employees. In doing so, leaders can influence employees’ values, beliefs and attitudes so that their performance exceeds the minimum requirements according to their employment contracts [24,26]. Transformational leadership remains a widely used concept in work and health research.

Given the requirements of the modern working world, rapid change due to increasing digitization, and unforeseen events (e.g., the SARS-CoV-II pandemic), psychological wellbeing and transformational leadership play a crucial role. Jamal et al. [18] found that employees’ wellbeing reduced stress in full-time telecommuters, and transformational leadership mitigated employee burnout during organizational change [27]. Most of the research on transformational leadership and psychological wellbeing is in line with the JD-R model and has concentrated on a leader-centric view that focusses on leadership style impacting employees’ psychological wellbeing. Several studies have revealed associations between transformational leadership and employees’ psychological wellbeing in different settings—for example, in nursing and health-care workers [28,29] or municipal employees [30]. A meta-analysis revealed strong associations between transformational leadership and positive mental health (e.g., wellbeing) [31]. In contrast, Eisele’s cross-sectional design [32] found no significant impact of transformational leadership on employees’ wellbeing and Nielsen [33] found no direct effect of transformational leadership on psychological wellbeing over time.

In addition to this leader-centric view, the follower-centric view has recently gained attention. The latter does not focus on leadership style influencing employee’s wellbeing but on employees’ mind-set being relevant for the evaluation of leadership behavior/style, which means that employee attitudes play a crucial role in the evaluation and acceptance of leaders, highlighting the relevance of employee characteristics in explaining their reactions [30,34,35]. Results of longitudinal studies indicate that employees’ wellbeing predicts transformational leadership [30,33]. However, Perko et al. [30] assert that transformational leadership and psychological wellbeing can be seen as antecedents for *“favourable or unfavourable development”*.

### 1.3. Research Question and Hypotheses

Even though the JD-R model is widely used and recommended, research on the reciprocal relationships between job characteristics and employees’ wellbeing is still needed [22]. Furthermore, although relevant research exists on the relationship between transformational leadership and psychological wellbeing, longitudinal studies on this topic are still rare and clarification is needed on how employees’ improving or declining wellbeing impacts on the perceived transformational leadership [30,36,37]. Research on this topic is also crucial for practitioners: in the case of a reciprocal relationship, organizations should focus on how leaders affect the psychological wellbeing of their employees while taking into account that employees’ psychological wellbeing might explain the positive or negative leadership feedback of employees [30].

Therefore, this study focuses on longitudinal data from two employee surveys in 2015 (t0) and 2018 (t1) in a medium-sized German company and examines the relationship between perceived transformational leadership and employees’ psychological wellbeing (see Figure 1). Considering the existing research and its partly contradictory results and based on the assumption of Perko et al. [30], the impact of declining or improving employees’ psychological wellbeing on transformational leadership and the impact of declining or improving perceived transformational leadership on employees’ psychological wellbeing is examined. Following the leader-centric view, we assume the following:

**H1.** 
*The difference in perceived transformational leadership over time significantly impacts psychological wellbeing at t1.*


Furthermore, following the follower-centric view, we assume that:

**H2.** 
*The difference in psychological wellbeing over time significantly impacts the leadership score at t1.*


## 2. Materials and Methods

This study builds on longitudinal data from an employee survey in a medium-sized German company in the lighting industry. Two surveys were conducted in 2015 and 2018 using paper-based questionnaires in 2015 and a web-based survey tool (LimeSurvey) in 2018. The surveys were aimed at all employees. In Germany, employers are subject to the Occupational Health and Safety Act and are obliged to identify and prevent or mitigate possible risks to the mental health of their employees [38]. The employee surveys were supported by the company, as such surveys comply with German employer obligations regarding mental risk assessment. Survey results in 2015 were used to reveal “red flags” in order to subsequently address them, for example via external support.

### 2.1. Study Design and Participants

All company employees (2015: N = 582; 2018: N = 537) were invited to participate in the surveys (total sample). Both surveys were supported by the company’s Human Resource Management by announcing the employee survey in the different departments and encouraging participation. In 2015, the employees received a paper-based questionnaire from their direct supervisors. Sealed collection boxes were available in the office to enable employees to anonymously return their questionnaires (sealed in an envelope). Employees who could not return the questionnaire personally due to illness or vacation received it by mail. An enclosed prepaid envelope could be sent directly to the evaluating institute (also anonymous). In 2018, the survey was conducted using a web-based survey tool. To ensure the survey was limited to company employees and that each person could only complete it once, all employees received from their direct supervisor an access key in a sealed envelope from the evaluating institute. This access key was known to the recipient alone. For employees with no access to a PC, rooms with PCs were provided. The survey took half an hour to complete, and employees were allowed to complete it during working hours. Study information was provided directly with the paper-based questionnaire in 2015, and participants had to confirm receipt of study information before being able to participate in the online survey in 2018 (via a PDF link).

Both surveys were available in German, and study participation was voluntary. Data were collected and analyzed anonymously. All participants provided consent and agreed with data analyses and scientific publications by the University of Cologne in anonymized form. The study was presented to and approved by the Ethics committee of the University of Cologne, Medical Faculty (application No. 20-1075).

Of the 582 employees in 2015, 408 answered and returned the questionnaire (response rate 70.1%). Of the 537 employees in 2018, 430 participated in the online survey (response rate 80.1%). To match individuals’ data over time, employees could voluntarily generate a personal code within the questionnaire (based on personal information such as “What is the first letter of your mother’s first name?”). With the help of personal codes in both surveys, we were able to link the data of 127 employees from 2015 (response rate 21.8%) to 2018 (response rate 23.6%) (see Figure 2). To include as many cases as possible in the analyses, missing values for all items were imputed (mean value of time series). The characteristics of the study sample are shown in Table 1.

### 2.2. Measures

To measure the **psychological wellbeing** of employees, we used the German version of the WHO (Five) Well-Being Index (WHO-5) questionnaire developed by the World Health Organization [39], which is the most widely accepted tool for measuring subjective psychological wellbeing. The WHO-5 comprises five questions focusing on individuals’ last two weeks—for example “Over the last two weeks I have felt cheerful and in good spirits.” All items were answered using a six-point scale from 0 (never) to 5 (the whole time). A raw value of 0 indicates the worst possible wellbeing, and 25 indicates the best possible wellbeing. Values below 13 can be seen as an indicator to test for depression [40]. Items were used in their original form. Cronbach’s Alpha was 0.854 in 2015 and 0.872 in 2018.

**Transformational leadership** was measured by six items following Podsakoff et al.—for example “Always gives me positive feedback when I perform well” [24,26]. In this study, all six items were in German and could be answered on a five-point Likert scale from 1 (never) to 5 (always). Cronbach’s Alpha in this study was 0.785 in 2015 and 0.753 in 2018.

As a confounding variable we referred to the validated SOCAPO-E instrument (**social capital**), which was aimed initially at employees in healthcare organizations [41]. It comprises six items based on Bauman’s concept of community [42] (including mutual understanding, warm circle, trust, sense of being a team, mutual help, and shared values)—for example “In our hospital, there is unity and agreement” [41]. In this study, *“In our hospital”* was replaced by *“In our company.”* All items could be answered on a four-point scale from 1 (strongly disagree) to 4 (strongly agree). In healthcare settings, the SOCAPO-E was associated with such factors as job satisfaction and burnout [41]. Cronbach’s Alpha was 0.867 in 2015 and 0.876 in 2018. Furthermore, age and gender were considered as confounding variables.

### 2.3. Data Analysis

We conducted paired *t*-tests to test for changes in transformational leadership, psychological wellbeing, and social capital over time. Multiple linear regression analyses were conducted to test whether (H1) differences (∆) in psychological wellbeing over time significantly impacted perceived transformational leadership at t1 and whether (H2) differences over time in perceived transformational leadership significantly impacted psychological wellbeing at t1.

In model I, we focused on transformational leadership_t1_ as the dependent variable and psychological wellbeing_∆_ as the independent variable. As confounding variables, we considered transformational leadership_t0_, social capital_∆_, age, and gender.

In model II, we used psychological wellbeing_t1t_ as the dependent variable and transformational leadership_∆_ as the independent variable. As confounding variables, we considered psychological wellbeing_t0_, social capital_∆_, age, and gender.

Before multiple linear regression analyses were conducted, we checked for multi-collinearity. Missing values were imputed by mean of each item, ranging from 1 to 7 imputations per item. All analyses were conducted using IBM SPSS Statistics 23.

## 3. Results

Of the study population, 65.4% were male and 33.1% were female, with 35.5% under and 62.2% 40 years and older (see Table 1). Social capital increased over time with t(126) = −3.985, *p* < 0.001. No significant decrease or increase was found for psychological wellbeing (t(126) = −0.546, *p* > 0.05) and perceived transformational leadership (t(126) = 1.343, *p* > 0.05). According to the cut-off score of 13 [40], employees showed inconspicuous average values in their psychological wellbeing in 2015 (14.85) and 2018 (15.09).

Before conducting multiple linear regression analyses, we checked for multi-collinearity. The results showed no multi-collinearity of variables (see Table 2 and Table 3).

Model I showed significant results for the determinant factors transformational leadership_t0_ (β = 0.548, *p* < 0.001) and gender (β = −0.283, *p* < 0.01). Psychological wellbeing_∆_ (β = 0.005, *p* > 0.05), social capital_∆_ (β = 0.134, *p* > 0.05), and age (β = 0.022, *p* > 0.05) had no significant impact on the perceived transformational leadership_t1_ (see Table 4).

In model II, significant results were revealed for the determinant factors psychological wellbeing_t0_ (β = 0.547, *p* < 0.001) and social capital_∆_ (β = 2.739, *p* < 0.01). Transformational leadership_∆_ (β = −0.119, *p* > 0.05), age (β = −0.761, *p* > 0.05) and gender (β = 1.074, *p* > 0.05) had no significant impact on psychological wellbeing_t1_ (see Table 5).

## 4. Discussion

In this study, changes in psychological wellbeing over time had no significant impact on perceived transformational leadership. Furthermore, changes in perceived transformational leadership over time had no significant impact on employees’ psychological wellbeing. Hence, we found no evidence to support the follower- or the leader-centric view and cannot support Hypotheses 1 and 2. However, the results are congruent with Nielsen [33], who found no direct impact of transformational leadership on psychological wellbeing, over time. Moreover, Eisele [32] found no predictive power for transformational leadership on employees’ wellbeing but found it for devious leadership. Regarding leadership behavior, it might be the case that devious, rather than transformational leadership, had a significant impact on employees’ psychological wellbeing in this longitudinal study.

Regarding psychological wellbeing, the study results indicate that perceived transformational leadership is not a significant factor, but the social climate in the company as a whole might play a more crucial role. This finding aligns with previous studies that found predictive validity of social capital (SOCAPO-E) on burnout and wellbeing [41]. Furthermore, this result supports the underlying assumptions of the JD-R model on the relationship between job characteristics and employees’ wellbeing.

Male employees in this study perceived better transformational leadership than female employees did. Previous studies found that the effectiveness of transformational leadership depends on the gender dyad of employee and supervisor, that male employees are more likely to be influenced by transformational leadership [43], and that male supervisors are more likely to be reported with higher levels of perceived transformational leadership [44]. In this study, we could not trace directly whether employees rate male or female supervisors, but most of the supervisors in the study company were men (>86%). It might be the case that men in this study were more likely to be influenced by transformational leadership and therefore perceived higher transformational leadership than their female colleagues. The fact that gender influenced transformational leadership in this study might slightly support the follower-centric view [30], as individual characteristics of employees were found to be relevant for perceived transformational leadership. However, it might not be gender itself that was relevant in this case, but underlying gender mechanisms. For example, referring to former studies, women might perceive a double burden of employment and household chores [45] and might also be confronted more frequently with lower rewards and fewer advancement opportunities as a result of the use of family-friendly policies (e.g., parental leave or part-time work) [46]. There is evidence that extended parental leave negatively impacts career and salary. Some managers might believe that employees using family-friendly policies—that are more often women—are less committed to the organization, have low career ambitions and perceive unfairness due to more work for supervisors and colleagues. Long parental leave can have negative effects on the future career and salary [47,48,49]. These aspects might result in different perceived transformational leadership in male and female workers.

It should be noted that both surveys were conducted before the outbreak of SARS-CoV-2. Regarding home-office work due to the pandemic, employees are now confronted with a new and challenging work environment (e.g., telecommunication) [18,19]. Furthermore, employees experienced challenges regarding organizational and leadership culture, including the negative image of home-office work, lack of trust on the part of managers and colleagues, and a high “presence culture” (the focus is less on the results achieved by employees, but more on workplace presence being seen as a sign of engagement) [50]. These factors could not be addressed in this study. The impact of home-office work on the relationship between transformational leadership and employees’ psychological wellbeing needs to be addressed in future research.

Nonetheless, the study results and the partly contradictory current state of research highlight the necessity of mental risk assessments in each company as underlying mechanisms regarding the psychological wellbeing of employees may differ between companies. Evidence-based mental risk assessments in the form of employee surveys are an appropriate tool for identifying individual risk factors for the psychological wellbeing of employees and should consider as many possible stressors and resources as possible. Based on the survey results, those responsible have indications for further procedures, e.g., implementing measures in cooperation with the employees to enhance the situation. Close cooperation with employees is essential for uncovering underlying mechanisms and implementing relevant company or departmental measures [51]. Rural-urban aspects should be considered when implementing workplace health promotion measures [52]. Furthermore, developments over time should be tracked and analyzed regularly.

### Strenghts and Limitations

In this study, we analyzed data from 127 employees. Therefore, the sample size was above the required number of 109 cases for conducting regression analyses with five independent variables, according to Green [53]. However, results on the relationship between transformational leadership and psychological wellbeing over time might require more cases or greater differences over time to reveal significant results.

The three-year gap between the surveys is relatively long, and due to linkage at the individual level, data for many employees were not included in this study. However, this was the only way of examining panel rather than trend data. In doing so, we were able to ensure that development over the three years was not related to different study samples (e.g., due to staff or employee changes) but to factors at the individual level. Nonetheless, staff changes might be relevant for changes over time (e.g., in social capital), affecting employees on an individual level. Furthermore, leadership-level staff changes might also be relevant for differences in employees’ perceived transformational leadership over time. In this study, most of the leadership positions were unchanged over time.

Limitations arising from social desirability and common method bias might have biased the study results. To address this possibility, employees were informed about the aim and the anonymity of the survey. Furthermore, the three-year gap might be beneficial in this case, as participants were unlikely to have remembered the answers, they gave three years earlier.

Furthermore, it is noteworthy that the perceived transformational leadership in this study sample was relatively high at both survey dates and that selection bias (e.g., through the healthy worker survivor effect [54]) cannot be totally excluded. The generalizability of the study results remains limited, as the study population was restricted to employees of a medium-sized German company in the lighting industry.

## 5. Conclusions

Even though we found no evidence of the impact of transformational leadership on psychological wellbeing over time, other studies have found associations between leadership style and the psychological wellbeing of employees. Therefore, companies should provide leaders with organizational resources (e.g., time for leadership tasks or training) to ensure good leadership behavior. Especially in the time of SARS-CoV-2 and the consequent rise in home-office work, employees perceived leadership behavior (e.g., lack of trust in employees) as challenging.

However, in this study sample, social capital and gender played a more crucial role in the psychological wellbeing of employees. The results highlight that it can be worthwhile for companies to invest in social capital and focus on gender aspects at work. As underlying mechanisms regarding the psychological wellbeing of employees may differ between companies, each organization should conduct mental risk assessments to identify “red flags” and implement suitable measures. Data should be tracked and analyzed over time.

Further longitudinal research on the relationship between psychological wellbeing and transformational leadership is needed and should focus on more confounding variables, always including social capital, gender, and SARS-CoV-2 factors.

## Figures and Tables

**Figure 1 ijerph-20-00676-f001:**
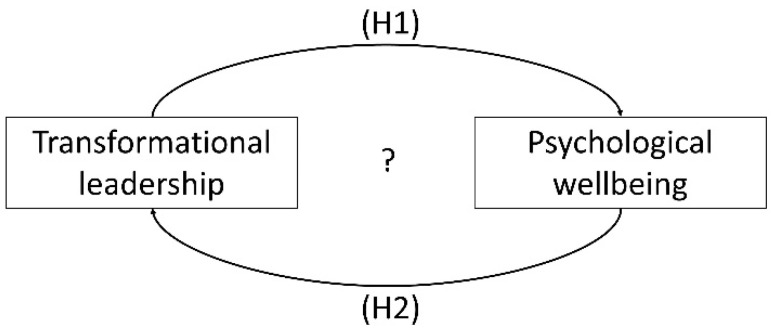
Conceptual model.

**Figure 2 ijerph-20-00676-f002:**
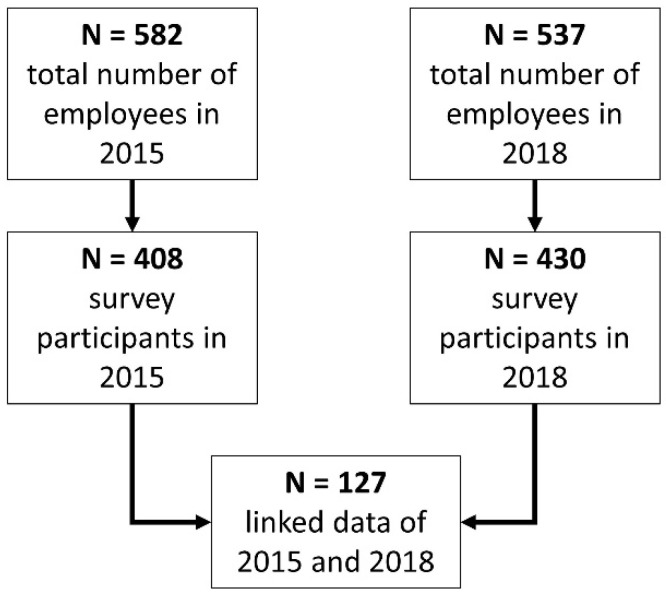
Flowchart of study sample.

**Table 1 ijerph-20-00676-t001:** Descriptions of study sample.

		N		%	
age					
	16–29 years	18		14.2	
	30 to 39 years	27		21.3	
	40 to 49 years	40		32.3	
	≥50 years	38		29.9	
	missing	3		2.4	
gender					
	Male	81		65.4	
	female	42		33.1	
	missing	2		1.6	
		**N**	**M**	**SD**	**Median (min/max)**
Transformational leadership_t0_		127	3.47	0.631	3.50 (1.33/4.67)
Transformational leadership_t1_		127	3.40	0.611	3.50 (1.67/4.50)
Psychological wellbeing_t0_		127	14.85	4.863	15.17 (2.00/25.00)
Psychological wellbeing_t1_		127	15.09	4.916	16.00 (1.00/25.00)
Social capital_t0_		127	2.60	0.536	2.67 (1.00/4.00)
Social capital_t1_		127	2.77	0.485	2.83 (1.67/4.00)

Notes: M = mean value; SD = standard deviation.

**Table 2 ijerph-20-00676-t002:** Correlations between variables, model I.

Variable	N	1	2	3	4	5	6
(1) Leadership_t0_	12	-	0.555 ***	−0.151	−0.247 **	−0.020	−0.070
(2) Leadership_t1_	123		-	0.041	−0.051	−0.005	−0.254 **
(3) Psychological wellbeing_∆_	123			-	0.349 ***	−0.075	0.134
(4) Social capital_∆_	123				-	0.054	0.026
(5) Age	123					-	0.013
(6) Gender	123						-

Notes: Pearson correlation r and α values (in the diagonal) are shown; ** *p* < 0.01; *** *p* < 0.001; Age was considered as bivariate variable.

**Table 3 ijerph-20-00676-t003:** Correlations between variables, model II.

Variable	N	1	2	3	4	5	6
(1) Psychological wellbeing_t0_	123	-	0.470 ***	−0.222 *	−0.239 **	0.038	−0.073
(2) Psychological wellbeing_t1_	123		-	−0.097	0.120	−0.039	0.065
(3) Leadership_∆_	123			-	0.213 *	−0.027	−0.190 *
(4) Social capital_∆_	123				-	0.054	0.026
(5) Age	123					-	0.013
(6) Gender	123						-

Notes: Pearson correlation r and α values (in the diagonal) are shown; * *p* < 0.05; ** *p* < 0.01; *** *p* < 0.001; Age was considered as bivariate variable.

**Table 4 ijerph-20-00676-t004:** Multiple linear regression analysis, model I.

Determinant Factors ^1^	Regression Coefficient B (SE)	Beta	*p* Value	95% Confidence Interval		R^2^ (Adjusted)
				Lower Value	Upper Value	
Transformational leadership_t0_	0.548 (0.073)	0.569	<0.001 ***	0.403	0.693	0.365 (0.338)
Psychological wellbeing_∆_	0.005 (0.010)	0.044	0.581	−0.014	0.024	
Social capital_∆_	0.134 (0.110)	0.098	0.225	−0.083	0.351	
Age ^2^	0.022 (0.094)	0.017	0.815	−0.164	0.208	
Gender	−0.283 (0.096)	−0.219	0.004 **	−0.473	−0.093	

Notes: SE = standard error; ** *p* < 0.01; *** *p* < 0.001. ^1^ dependent variable: leadership_t1_. ^2^ Age was, due to small group sizes, considered as bivariate variables (age group 1 = 16 to 39 years, group 2 = 40 to ≥ 50 years).

**Table 5 ijerph-20-00676-t005:** Multiple linear regression analysis, model II.

Determinant Factors ^1^	Regression Coefficient B (SE)	Beta	*p* Value	95% Confidence Interval		R^2^ (Adjusted)
				Lower Value	Upper Value	
Psychological wellbeing_t0_	0.547 (0.082)	0.540	<0.001 ***	0.384	0.710	0.298 (0.268)
Transformational leadership_∆_	−0.119 (0.693)	−0.014	0.864	−1.491	1.253	
Social capital_∆_	2.739 (0.897)	0.247	<0.003 **	0.963	4.514	
Age ^2^	−0.761 (0.796)	−0.074	0.341	−2.338	0.815	
Gender	1.074 (0.829)	0.103	0.198	−0.568	2.716	

Notes: SE = standard error; ** *p* < 0.01. *** *p* < 0.001. ^1^ dependent variable: psychological wellbeing_t1_. ^2^ Age was, due to small group sizes, considered as bivariate variables (age group 1 = 16 to 39 years, group 2 = 40 to ≥ 50 years).

## Data Availability

The data presented in this study are available on request from the corresponding author. The data are not publicly available due to data protection reasons.

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
