# Peer review of "Transformational Leadership and Employees’ Psychological Wellbeing: A Longitudinal Study"

_ijerph, 2022, doi:10.3390/ijerph20010676_

Round 1

Reviewer 1 Report

Comments to Authors

Recommendation: Reject

First, I would like to thank the Editor for trusting me with the opportunity to review this research, "Transformational leadership and employees’ psychological wellbeing: a longitudinal study." The theme of the study is interesting and has the potential to make a contribution; however, there are certain apprehensions and flaws which compelled me to warrant a rejection. The detailed comments are as below:

1- The introduction section starts well by building upon the background about the role of managerial/supervisory/organizational support and its impact on employees’ psychological wellbeing and performance. The authors have also referred to the JD-R Framework as well for drawing support regarding the positive and negative roles of job resources and demands on employees' wellbeing and psychological and mental strains. However, at the end of the introduction section, the study goes off-track and does not clearly come up with the research problem(s) it is exactly addressing and setting the objectives accordingly. The authors need to fine-tune the introduction signifying the objectives and the need of the study with more clarity. Moreover, the study is also under-referenced and rather cites older references in the introduction section. It needs to include some of the latest references. Some of the suggested references are as given below. Further, the authors have also referred to SARS-CoV-2 (Coronavirus) in the introduction as an influencing force on leadership, while the study period is from 2015 to 2018, which is before the Covid period. Therefore, it needs justification that how does Covid situation connect with the study.

Jamal, M.T., Anwar, I., Khan, N.A. and Saleem, I. (2021), "Work during COVID-19: assessing the influence of job demands and resources on practical and psychological outcomes for employees", Asia-Pacific Journal of Business Administration, https://doi.org/10.1108/APJBA-05-2020-0149

Hayes, S. W., Priestley, J. L., Moore, B. A., & Ray, H. E. (2021). Perceived Stress, Work-Related Burnout, and Working From Home Before and During COVID-19: An Examination of Workers in the United States. SAGE Open, 11(4), 21582440211058190.

Jamal, M. T., Alalyani, W. R., Thoudam, P., Anwar, I., & Bino, E. (2021). Telecommuting during COVID 19: A Moderated-Mediation Approach Linking Job Resources to Job Satisfaction. Sustainability, 13(20), 11449. https://doi.org/10.3390/su132011449

Kumar, P., Kumar, N., Aggarwal, P., & Yeap, J. A. L. (2021). Working in lockdown: The relationship between COVID-19 induced work stressors, job performance, distress, and life satisfaction. Current Psychology, 40(12), 6308–6323.

2- I suggest the authors separate the review and hypotheses development from the introduction section and make it a second section as “Literature and Hypotheses” or whatever suitable heading they find suitable.

3- As the study’s design is time-lagged longitudinal and the time lag between two waves of the data used in the study is way too long (three years), there are certain apprehensions that need justifications.

·       Why the time-lag between wave-1 and wave-2 was three years?

·       Was there any intervention/experimentation during the time-lag period to transform the perceived levels of transformational leadership among the employees as the study has measured the change (Δ) in the perceived levels of transformational leadership, psychological wellbeing, and social capital  (from T0 to T1).  

·       As the study adopts a time-lagged design, the authors should have reported how did they match the data from T1 to T2? There is no mentioning of any response ID creation by the respondents at T1 so that it could be used for the data matching at T2.

·       Time lag of three years is too long, and in the meantime, many employees (who responded to the T1 questionnaire) might have left their jobs or switched to another organization. Thus, it would be worth that mentioning how did the authors tackle this issue?

4- Since the study has used latent variables (with 19 questionnaire items), the final matched sample of the study is way too low as the rule of thumb for sample determination is a minimum of 10 responses per questionnaire item which accounts for a minimum sample of 190 final responses.

5- The study has not even established the measurement model (i.e., convergent validity and discriminant validity) through confirmatory factor analysis.

6- The results of the regression analyses do not seem plausible as the Lower and Upper values of Confidence Intervals do not match the calculation using the formula; LLCI/ULCI = SE × Z Score (i.e., 1.96 at 5% ) ± β

Where ‘LLCI’  is Lower limit confidence interval, ‘ULCI’ is Upper limit confidence interval, ‘SE’ is the standard error of the regression weight, and ‘β’ is the regression weight.

7-  Discussion is too weak and does not discuss the findings while conforming/critiquing the previous literature.

8- Even no suggestions, based on the findings, for the practical implications have been offered.

Author Response

Dear reviewer,

First of all thank you for reading our manuscript in detail and giving valuable feedback! Please find our point-by-point answers in the following.

However, at the end of the introduction section, the study goes off-track and does not clearly come up with the research problem(s) it is exactly addressing and setting the objectives accordingly. The authors need to fine-tune the introduction signifying the objectives and the need of the study with more clarity. Moreover, the study is also under-referenced and rather cites older references in the introduction section. It needs to include some of the latest references. Some of the suggested references are as given below. Further, the authors have also referred to SARS-CoV-2 (Coronavirus) in the introduction as an influencing force on leadership, while the study period is from 2015 to 2018, which is before the Covid period. Therefore, it needs justification that how does Covid situation connect with the study.

Jamal, M.T., Anwar, I., Khan, N.A. and Saleem, I. (2021), "Work during COVID-19: assessing the influence of job demands and resources on practical and psychological outcomes for employees", Asia-Pacific Journal of Business Administration, https://doi.org/10.1108/APJBA-05-2020-0149

Hayes, S. W., Priestley, J. L., Moore, B. A., & Ray, H. E. (2021). Perceived Stress, Work-Related Burnout, and Working From Home Before and During COVID-19: An Examination of Workers in the United States. SAGE Open, 11(4), 21582440211058190.

Jamal, M. T., Alalyani, W. R., Thoudam, P., Anwar, I., & Bino, E. (2021). Telecommuting during COVID 19: A Moderated-Mediation Approach Linking Job Resources to Job Satisfaction. Sustainability, 13(20), 11449. https://doi.org/10.3390/su132011449

Kumar, P., Kumar, N., Aggarwal, P., & Yeap, J. A. L. (2021). Working in lockdown: The relationship between COVID-19 induced work stressors, job performance, distress, and life satisfaction. Current Psychology, 40(12), 6308–6323.

The introduction section was fundamentally revised. To avoid the impression that our study addresses SARS-CoV-2 aspects, we do no longer refer to this in the introduction section and moved this part to the discussion. Furthermore, we included current literature.

2- I suggest the authors separate the review and hypotheses development from the introduction section and make it a second section as “Literature and Hypotheses” or whatever suitable heading they find suitable.

We now use different headings in the introduction section, including “hypotheses”.

3- As the study’s design is time-lagged longitudinal and the time lag between two waves of the data used in the study is way too long (three years), there are certain apprehensions that need justifications.

  • Why the time-lag between wave-1 and wave-2 was three years?
  • Was there any intervention/experimentation during the time-lag period to transform the perceived levels of transformational leadership among the employees as the study has measured the change (Δ) in the perceived levels of transformational leadership, psychological wellbeing, and social capital (from T0 to T1). 

Thank you for your comment. It now says: “The employee surveys were supported by the company itself as, with the surveys, it complies with German employer obligations regarding the mental risk assessment. Survey results in 2015 were used to reveal “red flags” in order to subsequently address them, e.g. with external support.”

  • As the study adopts a time-lagged design, the authors should have reported how did they match the data from T1 to T2? There is no mentioning of any response ID creation by the respondents at T1 so that it could be used for the data matching at T2.

It now says “To match data of individuals over time, employees could voluntarily generate a personal code within the questionnaire (based on personal information like “What is the first letter of your mother's first name?”). With the help of the personal code in both surveys we were able to link the data of 127 employees from 2015 (response rate 21.8%) to 2018 (response rate 23.6%) (see figure 2).”

  • Time lag of three years is too long, and in the meantime, many employees (who responded to the T1 questionnaire) might have left their jobs or switched to another organization. Thus, it would be worth that mentioning how did the authors tackle this issue?

We address this in the Strenghts and limitations section. It now says “The three-year-gap between both surveys is relatively long and due to linkage at individual level, data of many employees were not included in this study sample. However, this was the only possibility to examine panel data instead of trend data. Doing so, we were able to ensure that the development over the three years was not related to different study samples (, e.g. due to staff or employee changes) but to aspects at individual level. Nonetheless, staff changes might be relevant for changes over time (, e.g. in social capital), affecting employees on an individual level. Furthermore, staff changes on leadership level might also be relevant for differences in employees’ perceived transformational leadership over time.”

4- Since the study has used latent variables (with 19 questionnaire items), the final matched sample of the study is way too low as the rule of thumb for sample determination is a minimum of 10 responses per questionnaire item which accounts for a minimum sample of 190 final responses.

We now imputed missing values for all relevant items. To address sample size it now says “In this study we were able to analyze data of 127 employees. Therefore, the sample size is above the required number of 109 cases for conducting regression analyses with five independent variables according to Green [51].”

5- The study has not even established the measurement model (i.e., convergent validity and discriminant validity) through confirmatory factor analysis.

According to sample size and as data is based on only one sample, we did not include validation aspects in this study.

6- The results of the regression analyses do not seem plausible as the Lower and Upper values of Confidence Intervals do not match the calculation using the formula; LLCI/ULCI = SE × Z Score (i.e., 1.96 at 5% ) ± β, Where ‘LLCI’  is Lower limit confidence interval, ‘ULCI’ is Upper limit confidence interval, ‘SE’ is the standard error of the regression weight, and ‘β’ is the regression weight.

We are sorry, the SE was missing in the tables and now inserted.

7-  Discussion is too weak and does not discuss the findings while conforming/critiquing the previous literature.

We revised the discussion and discussed findings against the backdrop of previous literature.

8- Even no suggestions, based on the findings, for the practical implications have been offered.

Thank you for your comment. It now says “It has to be considered that both surveys were conducted before the outbreak of SARS-CoV-2. Regarding home office as a result of the pandemic, employees nowadays are confronted with a new and partly challenging work environment (, e.g. telecommunica-tion) [9,10]. Furthermore, employees experienced challenges with regard to organizational and leadership culture, e.g. negative image of home office, lack of trust on the part of managers and colleagues and/or a high “presence culture” (focus is less on the results achieved by employees, but rather is workplace presence seen as a sign of engagement) [48]. This could not be addressed in this study. The impact of home office on the relation-ship between (transformational) leadership and employees’ psychological wellbeing needs to be addressed in future research. Nonetheless, study results and the partly contradictory current state of research in general highlight the necessity of mental risk assessments in each company as underlying mechanisms regarding the psychological wellbeing of employees may differ between companies. Evidence-based mental risk assessments in form of employee surveys are an appropriate tool to identify individual risk factors for the psychological wellbeing of em-ployees and should consider as many possible stressors and resources as possible. Based on the survey results, those responsible have indications for further procedures, e.g. im-plementing measures in cooperation with the employees to enhance the situation. The close cooperation with employees is particularly important in order to uncover underlying mechanisms and to implement company and/or department relevant measures [49]. When implementing workplace health promotion measures, rural-urban aspects should be considered [50]. Furthermore, developments over time should be tracked and analyzed on a regular basis.”

Reviewer 2 Report

Abstract: It is not clear why the authors prefered a longitudinal design for this study? How this design sparks, already existing knowledge? Lastly, please remove the statistical results like beta and p values from abstract. Try to keep your abstract as simple as it could be for a better understanding and interest of the readers.

Introduction: It is not clear why different variables, for example, psychological wellbeing, were chosen in a transformational leadership framework?

Reference required for the below statement

Particularly against the backdrop of demographic change and the associated shortage of skilled workers, the mental health of employees is also of great importance from an economic point o……..”

According to the job-demand-resource model there are several resources and demands at work that may influence employees’ psychological wellbeing

Indicate the origin of this job-demand-model and indicate why it is important to relate it with this work.

Leadership style is associated with psychological wellbeing of employees and may function either as resource or stressor”

How wellbeing maybe a stressor?

Most of the research so far has concentrated on a leader-centric view, that focusses on leadership (style) having an impact on the psychological wellbeing of employees. In later research, the follower-centric view is gaining attention. This view does not focus on the leadership style influencing the wellbeing of employees but on employees actual mind-set being relevant for the evaluation of leadership behavior/style [10].

Why is it important to subscribe to a follower centric view?

The objectives of this study should be clearly specified in introduction part

There is no evidence provided in introduction why this study was important, and how it advances the already existing literary debate?

 Moreover, the rational, gap, motivation to carry out this research were not highlighted in introduction.

Introduction is a main part of your document which should be written with a due care. Kindly revise to better grab the attention of the readers. Thanks

Material and method

Please produce a conceptual model of your research so that the hypothesized relations should be more obvious to the readers.

How were the participants chosen?

The ethical statement is missing

What was the response rate?

How the issues like social desirability and common method bias addressed?

Measures:

Please explicitly state the origin of your scales. Moreover, indicate whether the items were adapted.

Discussion: This is not the way you presented this segment. Kindly first discuss the results of your study and then compare it to previous studies. Also, align these results with your study objectives.

Moreover, what are the implications for theory and practice?

Remove references from the conclusion part please. 

Author Response

Dear reviewer,

First of all thank you for reading our manuscript in detail and giving valuable feedback! Please find our point-by-point answers in the following.

Abstract: It is not clear why the authors prefered a longitudinal design for this study? How this design sparks, already existing knowledge?

We revised the abstract, where it now says: “Several studies already revealed associations between transformational leadership and psychological wellbeing of employees in different settings, while others did not. As research based on longitudinal data still is rather rare, this study builds on longitudinal data of two employee surveys in 2015 and 2018 in a German medium-sized company.”

Lastly, please remove the statistical results like beta and p values from abstract. Try to keep your abstract as simple as it could be for a better understanding and interest of the readers.

removed

Introduction: It is not clear why different variables, for example, psychological wellbeing, were chosen in a transformational leadership framework?

We fundamentally revised the introduction section to clarify this issue.

Reference required for the below statement

“Particularly against the backdrop of demographic change and the associated shortage of skilled workers, the mental health of employees is also of great importance from an economic point o……..”

Reference added.

“According to the job-demand-resource model there are several resources and demands at work that may influence employees’ psychological wellbeing”

Reference added.

Indicate the origin of this job-demand-model and indicate why it is important to relate it with this work.

We added more information on the JD-R model in the introduction section. It now says “One theoretical model, that gives a conceptual framework on which work aspects may be positively or negatively associated with employees’ psychological wellbeing is the job-demands-resource model (JD-R model) [8]. The JD-R model is widely used by re-searchers (, e.g. [9–17]) and helped eliminating some limitations of previous theoretical models (, e.g. the job-demand-control model of Karasek [18]) [19]. The authors of a me-ta-analytic review based on longitudinal studies conclude that the JD-R model provides a valuable theoretical framework regarding employees’ wellbeing [20].”

Leadership style is associated with psychological wellbeing of employees and may function either as resource or stressor”

This sentence was deleted.

 “Most of the research so far has concentrated on a leader-centric view, that focusses on leadership (style) having an impact on the psychological wellbeing of employees. In later research, the follower-centric view is gaining attention. This view does not focus on the leadership style influencing the wellbeing of employees but on employees actual mind-set being relevant for the evaluation of leadership behavior/style [10]”.

Why is it important to subscribe to a follower centric view?

We addressed this in the introduction section. It now says “Most of the research so far is in line with the job-demand-resources-model and has concentrated on a leader-centric view, that focusses on leadership (style) having an impact on the psychological wellbeing of employees. Several studies already revealed associa-tions between transformational leadership and psychological wellbeing of employees in different settings, e.g. in nursing and health-care workers [25,26] or municipal employees [27]. Also, a meta-analysis revealed strong associations between transformational leader-ship and positive mental health, e.g. wellbeing [28]. Furthermore, transformational lead-ership is perceived to mitigate burnout of employees during organizational change [29]. Contrary to this, Eisele [30] found no impact of transformational leadership on employees’ wellbeing in a cross-sectional design and Nielsen [31] found no direct impact of transfor-mational leadership on psychological wellbeing over time.”

The objectives of this study should be clearly specified in introduction part, There is no evidence provided in introduction why this study was important, and how it advances the already existing literary debate? Moreover, the rational, gap, motivation to carry out this research were not highlighted in introduction. Introduction is a main part of your document which should be written with a due care. Kindly revise to better grab the attention of the readers. Thanks

Thank you for your comment. We clarified the objective and the relevance of this study. It now says “Although there is already some relevant research on the relationship between trans-formational leadership and psychological wellbeing, longitudinal studies on this topic still are rather rare and it needs clarification on how improving or declining wellbeing of employees impacts on the perceived transformational leadership [27,34,35]. Furthermore, even though the JD-R model is widely used and recommended, research on the reciprocal relationships between job characteristics and employees’ wellbeing is still needed [20]. The aim of this study is to address this research gap and contribute to the current state of research by focusing on longitudinal data of two employee surveys in 2015 (t0) and 2018 (t1).”

Material and method

Please produce a conceptual model of your research so that the hypothesized relations should be more obvious to the readers.

Added, see figure 1.

How were the participants chosen?

The survey was aimed at all employees of the company.

The ethical statement is missing

We referred to the ethical statement in the “study design and participants” section. It says “The study was presented to and approved by the Ethics committee of the University of Cologne, Medical Faculty (application No. 20-1075).”

What was the response rate?

Response rate for linked data was added. It now says “Of the 582 employees in 2015, 408 answered and returned the questionnaire (re-sponse rate 70.1%). Of the 537 employees in 2018, 430 participated in the online survey (response rate 80.1%). To match data of individuals over time, employees could voluntar-ily generate a personal code within the questionnaire (based on personal information like “What is the first letter of your mother's first name?”). With the help of the personal code in both surveys we were able to link the data of 127 employees from 2015 (response rate 21.8%) to 2018 (response rate 23.6%) (see figure 2).”

How the issues like social desirability and common method bias addressed?

Thank you for your comment. We added this in the “strengths and limitations” section. It now says “Lastly, limitations through social desirability and common method bias might have biased study results. To address this issues, employees were informed about the aim and the anonymity of the survey. Furthermore, the three-year-gap might be beneficial in this case, as participants likely not remembered the answers they gave three years earlier.”

Measures:

Please explicitly state the origin of your scales. Moreover, indicate whether the items were adapted.

We added information for each measure.

Discussion: This is not the way you presented this segment. Kindly first discuss the results of your study and then compare it to previous studies. Also, align these results with your study objectives.

Thank you. We revised the discussion according to your comment.

Moreover, what are the implications for theory and practice?

We now discussed this in different parts of the discussion, e.g.:

“Furthermore, this result supports the underlying assumptions of the JD-R model on the relationships between job characteristics and employees’ wellbeing.”

“The impact of home office on the relationship between (transformational) leadership and employees’ psychological wellbeing needs to be addressed in future research.”

“Nonetheless, study results and the partly contradictory current state of research in general highlight the necessity of mental risk assessments in each company as underlying mechanisms regarding the psychological wellbeing of employees may differ between companies. Evidence-based mental risk assessments in form of employee surveys are an appropriate tool to identify individual risk factors for the psychological wellbeing of em-ployees and should consider as many possible stressors and resources as possible. Based on the survey results, those responsible have indications for further procedures, e.g. im-plementing measures in cooperation with the employees to enhance the situation. The close cooperation with employees is particularly important in order to uncover underlying mechanisms and to implement company and/or department relevant measures [49]. When implementing workplace health promotion measures, rural-urban aspects should be considered [50]. Furthermore, developments over time should be tracked and analyzed on a regular basis.”

Remove references from the conclusion part please.

Removed.

Reviewer 3 Report

I acknowledge this work. I will insist only on some issues that need further changes:

·         I noticed the lack of contributions in the Introduction section. It would be informative to address theoretical, methodological and practical contributions.

·         While the authors conceptualized transformational leadership, they do not provide a conceptualization for psychological well-being.

·         Beta should be used as symbol in the Abstract and Results section.

·         The theoretical implications of the findings should be presented.

Author Response

Dear reviewer,

First of all thank you for reading our manuscript and giving valuable feedback! Please find our point-by-point answers in the following.

  • I noticed the lack of contributions in the Introduction section. It would be informative to address theoretical, methodological and practical contributions.

Thank you for your comment. We fundamentally revised the introduction part that should now address those issues.

  • While the authors conceptualized transformational leadership, they do not provide a conceptualization for psychological well-being.

We added a conceptualization for psychogical wellbeing.

  • Beta should be used as symbol in the Abstract and Results section.

Done.

  • The theoretical implications of the findings should be presented.

We fundamentally revised the discussion where it, for example, now says “Furthermore, this result supports the underlying assumptions of the JD-R model on the relationships between job characteristics and employees’ wellbeing.”

“Therefore, we found no evidence to support the follower- nor the leader-centric view and cannot support hypotheses 1 and 2 in this study.”

Round 2

Reviewer 1 Report

First, I commend the authors for putting in substantial efforts in revising the manuscript. The authors have attempted to address all the comments with due rigor and have been successful to a greater extent. However, I find the introduction has become too short and does not serve the purpose of why this study is needed and what specific research problems it addresses. The authors need to work on the introduction section and improve significantly.

Author Response

Dear Reviewer,

first of all many thanks for the recognition of our work! Your comments helped improving the manuscript signifcantly.

Regarding your latest comment, we now revised the introduction section by adding new information and by revising formulations to highlight the purpose of why this study is needed and which research gaps it addresses more strongly.

Parts we added were:

"Especially regarding the requirements of the modern working world, the rapid change due to increasing digitization and in case of unforeseen events, e.g. the SARS-CoV-II pandemic, psychological wellbeing and transformational leadership play a crucial role. Jamal et. al [18] found that employees’ wellbeing reduced stress in full-time telecommuters and transformational leadership is perceived to mitigate burnout of employees during organizational change [27]. Most of ..."

AND

"Research on this topic is also of importance for practitioners: in case of a reciprocal relationship, organizations should not only focus on leaders affecting the psychological wellbeing of their employees, but also taking into account, that the employees’ psychological wellbeing as such might explain positive or negative leadership feedbacks of employees [30]."

Reviewer 2 Report

The revised manuscript looks great. 

I am happy to accept it in its current form. 

Thanks!

Author Response

Thank you very much for accepting our manuscript!